# Global Geometry of Bayesian Statistics

**DOI:** 10.3390/e22020240

**Published:** 2020-02-20

**Authors:** Atsuhide Mori

**Affiliations:** Department of Mathematics, Osaka Dental University, Osaka 573-1121, Japan; mori-a@cc.osaka-dent.ac.jp

**Keywords:** information geometry, poisson structure, symplectic structure, contact structure, foliation, Cholesky decomposition

## Abstract

In the previous work of the author, a non-trivial symmetry of the relative entropy in the information geometry of normal distributions was discovered. The same symmetry also appears in the symplectic/contact geometry of Hilbert modular cusps. Further, it was observed that a contact Hamiltonian flow presents a certain Bayesian inference on normal distributions. In this paper, we describe Bayesian statistics and the information geometry in the language of current geometry in order to spread our interest in statistics through general geometers and topologists. Then, we foliate the space of multivariate normal distributions by symplectic leaves to generalize the above result of the author. This foliation arises from the Cholesky decomposition of the covariance matrices.

## 1. Introduction

Suppose that a smooth manifold *U* is embedded in the space of positive probability densities defined on a fixed domain. Then, the relative entropy defines a separating premetric $D:U\times U\to R_{\geq 0}$ on *U*. Here a premetric on *U* is a non-negative function on U×U vanishing along the diagonal set Δ⊂U×U, and it is separating if it vanishes only at Δ. Its jet of order 3 at Δ induces a family of differential geometric structures on *U*, which is the main subject of the information geometry. There is a large body of literature on the information geometry (see [1,2] and references therein). It is worth noting that another “canonical” choice of premetric other than the above *D* is discussed in [3].

In the case where *U* is the space of univariate normal distributions, the half plane $H=R\times R_{>0}∋\left(m,s\right)$ presents *U*, where *m* denotes the mean and *s* the standard deviation. Since the convolution of two normal densities is a normal density, it induces a product ∗ on H called the convolution product. On the other hand, since the pointwise product of two normal densities is proportional to a normal density, it induces another product · on H called the Bayesian product. Their expressions are
(1)$$\left(m,s\right)*\left(m^{\prime },s^{\prime }\right)=\left(m+m^{\prime },\sqrt{s^{2}+{s^{\prime }}^{2}}\right),\;\left(m,s\right)\cdot \left(m^{\prime },s^{\prime }\right)=\left(\frac{m{s^{\prime }}^{2}+m^{\prime }s^{2}}{s^{2}+{s^{\prime }}^{2}},\frac{ss^{\prime }}{\sqrt{s^{2}+{s^{\prime }}^{2}}}\right).$$

In the previous work of the author ([4]), a Fourier-like transformation is defined as the diffeomorphism $\hat{F}:H\to H$ sending $\left(m^{\prime },s^{\prime }\right)$ to $\left(M,S\right)=\left(-\frac{m^{\prime }}{{s^{\prime }}^{2}},\frac{1}{s^{\prime }}\right)$. It is an involution interchanging the two operations ∗ and ·. Accordingly, the stereograph $f:H\times H\to R_{>0}$ of the above *D* is defined by
(2)$$\begin{array}{cc} f:(m,s,M,S)\mapsto & \frac{1}{2}{\left(\frac{M}{S}+sS\frac{m}{s}\right)}^{2}+\frac{1}{2}{\left(sS\right)}^{2}-\frac{1}{2}-\ln\left(sS\right)\\ \; & =\frac{1}{2}\left\{\frac{{\left(m-m^{\prime }\right)}^{2}+s^{2}}{{s^{\prime }}^{2}}-1+\ln\frac{s^{2}}{{s^{\prime }}^{2}}\right\}=D\left(\left(m,s\right),\left(m^{\prime },s^{\prime }\right)\right). \end{array}$$
The flow $\left(m,s,M,S\right)\mapsto \left(e^{t}m,e^{t}s,e^{-t}M,e^{-t}S\right)$ (t∈R) preserves *f* as well as the graph F⊂H×H of $\hat{F}$. The same symmetry appears in the contact/symplectic geometry related to the algebraic geometry of Hilbert modular cusps. Moreover, there exists a contact Hamiltonian flow whose restriction to the graph *F* presents a certain Bayesian inference. Its application appears in [5].

In this paper, we describe Bayesian statistics in the language of current geometry in order to share the problems among general geometers and topologists. Then, generalizing the above result of the author, we foliate the space of multivariate normal distributions by using the Cholesky decomposition of the covariance matrices and define on each leaf the Fourier-like transformation, the stereograph of the relative entropy, and the contact Hamiltonian flow presenting a Bayesian inference. The ultimate aim of this research is to construct a Bayesian statistical model of space-time on which everything learns by changing its inner distribution along the leaf.

## 2. Results

### 2.1. Symplectic/Contact Geometry

Current geometry does not heavily use tensor calculus. Instead, it uses (exterior) differential forms, which can be integrated along cycles, pulled-back under smooth maps, and differentiated without affine connections. In symplectic/contact geometry, the readers must be familiar with differential forms. Then, this subsection is the minimal summary of definitions. For the details, refer to [6].

A (positive) symplectic form on an oriented 2n-manifold is a closed 2-form ω satisfying $\omega^{n}>0$, where $\omega^{n}=\omega \wedge \cdots \wedge \omega$. If the orientation is reversed, the 2-form ω becomes a negative symplectic form. In either case, a symplectic form ω identifies a vector field *X* with an exact 1-form dH through the one-to-one correspondence defined by Hamilton’s equation $\iota_{X}\omega =-dH$. Here ι denotes the interior product. Then, *X* is called a Hamiltonian vector field of the primitive function *H* (+constant). The flow generated by *X* preserves the symplectic form ω. Namely, the Lie derivative $L_{X}\omega \left(=\iota_{X}d\omega +d\iota_{X}\omega \right)$ vanishes. A Lagrangian submanifold is an *n*-manifold which is immersed in a symplectic 2n-manifold so that the pull-back of the symplectic form vanishes. The word “symplectic” is a calque of “complex”. Indeed, there exists an almost complex structure *J* which is compatible with a given symplectic structure, i.e., for which the composition ω(·,J·) is a Riemannian metric. In the case where *J* is integrable, ω is called a Kähler form of the complex manifold.

On the other hand, a (positive) contact form on an oriented (2n−1)-manifold *N* is a 1-form η satisfying $\eta \wedge {\left(d\eta \right)}^{n-1}>0$. A (co-oriented) contact structure on *N* is the conformal class of a contact form. It can be presented as the oriented hyperplane field kerη. The product manifold R(∋t)×N carries the exact symplectic form $d(e^{t}\eta )$. Take a function *h* on *N*. Let *X* be the Hamiltonian vector field of the function $e^{t}h$ defined on the product manifold R×N. Then, the push-forward *Y* of *X* under the projection of R×N to the second factor is well-defined. The vector field *Y* is called the contact Hamiltonian vector field of the function *h* on *N*. The pair of the equations η(Y)=h and $\eta \wedge L_{Y}\eta =0$ uniquely determines *Y*. A Legendrian submanifold is an (n−1)-manifold which is immersed in a contact (2n−1)-manifold so that the pull-back of the contact form vanishes.

### 2.2. Bayesian Statistics

Suppose that any point *y* of a smooth manifold *M* equipped with a volume form dvol presents a positive probability density or probability $\rho_{y}:W\to R_{>0}$ defined on a (possibly discrete) measurable space *W*, where $\rho_{y}$ depends smoothly on *y*, and $\rho_{y}\neq \rho_{y^{\prime }}$ for $y\neq y^{\prime }\in M$. Let V be the space of positive volume forms with finite total volume on *M*. Take an arbitrary element V∈V and consider it as the initial state of the mind *M* of an agent. Here *W* stands for (a part of) the world for the agent. Finding a datum w∈W in his world, the agent can take the value $\rho_{y}\left(w\right)$ as a smooth positive function ${\bar{\rho }}_{w}:y\mapsto \rho_{y}\left(w\right)$ on *M*, which is called the likelihood of the datum *w*. Then, he can multiply the volume form *V* by ${\bar{\rho }}_{w}>0$ to obtain a new element of V. This defines the updating map
(3)$$\varphi :W\times V∋\left(w,V\right)\mapsto {\bar{\rho }}_{w}V\in V.$$
The “psychological” probability density $p_{V}$ on the mind *M* defined by $p_{V}dvol=V/\int_{M}V$ is accordingly updated into the density $p_{\varphi (w,V)}\propto p_{V}{\bar{\rho }}_{w}$, which is called the conditional probability density given the datum *w*. Practically, Bayes’ rule on conditional probabilities is expressed as
(4)$$P\left(y\in \Delta_{y}\;|\;w\in \Delta_{w}\right)=\frac{P(w\in \Delta_{w}\;|\;y\in \Delta_{y})}{P(w\in \Delta_{w})}\cdot P\left(y\in \Delta_{y}\right).$$
Here *P* denotes the probability of an event, $\Delta_{y}$ (respectively, $\Delta_{w}$) a small portion of *M* (respectively, *W*). Since the state of the world does not depend on the mind of the agent, the probability $P(w\in \Delta_{w})$ is independent of *y*, and therefore approximates a constant on *M*. On the other hand, the conditional probability $P(w\in \Delta_{w}\;|\;y\in \Delta_{y})$ of the datum *w* approximates a function of *y* which is clearly proportional to the above likelihood. This implies that the factor $\frac{P(w\in \Delta_{w}\;|\;y\in \Delta_{y})}{P(w\in \Delta_{w})}$ in the right-hand side of Bayes’ rule (Equation (Equation 4)) is approximately proportional to the likelihood. Thus, Bayes’ rule (Equation (Equation 4)) implies the updating of $p_{V}$ via the formula (Equation (Equation 3)). The Bayesian product · mentioned in the introduction appears in this context. Namely, the variable of the first factor is the mean *y* of a normal distribution on *W*. The density of the normal distribution at the datum *w* can be considered as a function of *y*, which is proportional to a normal distribution on *M*. Thus the Bayesian product of normal distributions on *M* presents the updating of the density of the predictive mean in the mind of the agent.

The aim of Bayesian updating is practical to many people. Indeed, the aim of the above updating is the estimation of the mean. Nevertheless, it is quite natural that a geometer multiplies a volume form by a positive function once he is given them. In this regard, we can say that the aim of Bayesian updating is a geometric setting of a dynamical system. In particular, a Bayesian updating in the conjugate prior is at first, simply the iteration of a Bayesian product.

### 2.3. The Information Geometry

Suppose that a manifold *U* is embedded in the subset $\left\{V\in V|\int_{M}V=1\right\}=\left\{p_{V}dvol|V\in V\right\}$. Hereafter, we identify the element $p_{V}dvol\in U$ with the “psychological” probability density $p_{V}$. We call *U* a conjugate prior for the updating map φ if the cone $\tilde{U}=\left\{e^{t}V|t\in R,V\in U\right\}$ satisfies $\varphi \left(W\times \tilde{U}\right)\subset \tilde{U}$. (Whether there exists a preferred conjugate prior or not, how to determine the initial state of the mind is another interesting problem. For example, one may fix the asymptotic behavior of the state of mind according to the aim of the Bayesian inference and search for the optimal decision of the initial state. See [7] for an approach to this problem via the information geometry.)

Now we define the “distance” $\tilde{D}:\tilde{U}\times \tilde{U}\to R$ on $\tilde{U}$, which satisfies none of the axioms of distance, by
(5)$$\tilde{D}\left(V_{1},V_{2}\right)=\int_{M}V_{1}\ln\frac{V_{1}}{V_{2}}\;\left(\mathrm{the}\mathrm{relative}\mathrm{entropy}\right).$$
From the convexity of −ln, we see that the restriction $D=\tilde{D}{|}_{U\times U}\left(\geq -\ln\left(\int_{M}V_{2}\right)=0\right)$ is a separating premetric on *U*, which is called the Kullback–Leibler divergence in information theory. This implies that the germ of *D* along the diagonal set Δ of U×U represents the zero section of the cotangent sheaf of *U*, that is, for any point $x=(x^{1},\cdots ,x^{n})$ of any chart of *U*, the Taylor expansion of $D\left(x+\frac{1}{2}dx,x-\frac{1}{2}dx\right)$ has no linear terms. Thus the differential dD:TU×TU→R also vanishes on the diagonal set $\Delta^{\prime }$ of TU×TU. We regard the 1-form on TU represented by the germ of dD along $\Delta^{\prime }$ as a quadratic tensor, and denote it by *g* (note that $g_{x}:T_{x}U\times T_{x}U\to R$ is linear). It appears as two times the quadratic terms $\frac{1}{2}\sum_{i,j}g_{ij}dx^{i}dx^{j}$ ($g_{ij}=g_{ji}$) in the above Taylor expansion. Of course, it also appears in the Taylor expansion of $D\left(x-\frac{1}{2}dx,x+\frac{1}{2}dx\right)$. Thus it can also be considered as the quadratic terms of the symmetric sum $D\left(x+\frac{1}{2}dx,x-\frac{1}{2}dx\right)+D\left(x-\frac{1}{2}dx,x+\frac{1}{2}dx\right)$. The symmetric matrix $\left[g_{ij}\right]$ is called the Fisher information in information theory. From the non-negativity of *D*, we may assume generically that *g* is a Riemannian metric. We would like to notice that this construction of Riemannian metric by means of symmetric sum also works over $\tilde{U}$. Indeed, we have
(6)$$\tilde{D}\left(e^{t+\frac{1}{2}dt}V,e^{t-\frac{1}{2}dt}V\right)+\tilde{D}\left(e^{t-\frac{1}{2}dt}V,e^{t+\frac{1}{2}dt}V\right)=e^{t}dt^{2}+O\left(dt^{3}\right).$$
Let $\nabla^{0}$ be the Levi–Civita connection of *g*. We write the lowest degree terms in the Taylor expansions of $D\left(x+\frac{1}{2}dx,x-\frac{1}{2}dx\right)-D\left(x-\frac{1}{2}dx,x+\frac{1}{2}dx\right)$ as $\frac{1}{3}\sum_{i,j,k}T_{ijk}dx^{i}dx^{j}dx^{k}$ ($T_{ijk}=T_{jik}=T_{ikj}$). This presents the symmetric cubic tensor *T*, which can be constructed from the anti-symmetric difference $D^{\prime }\left(x_{1},x_{2}\right)=D\left(x_{1},x_{2}\right)-D\left(x_{2},x_{1}\right)$ similarly as above. One can use it to deform the Levi–Civita connection $\nabla^{0}$ into the α-connections $\nabla^{\alpha }=\nabla^{0}-\alpha g^{*}T$ (α∈R) without torsion, where $g^{*}$ denotes the dual metric. Especially, we call $\nabla^{1}$ and $\nabla^{-1}$ respectively the e-connection and the m-connection. The symmetric tensor *T* is sometimes called skewness since it presents the asymmetry of *D*. The information geometry concerns the family of α-connections as well as the Fisher information metric on *U*. We usually do not extend it over $\tilde{U}$ for the symmetric sum of $\tilde{D}$ lacks asymmetry.

### 2.4. The Geometry of Normal Distributions

We consider the space *U* of multivariate normal distributions on $M=R^{n}$. A vector $\mu ={\left(\mu_{i}\right)}_{1\leq i\leq n}\in R^{n}$ and an upper triangular matrix $C={\left[c_{ij}\right]}_{1\leq i,j\leq n}\in \mathrm{Mat}\left(n,R\right)$ with positive diagonal entries parameterize *U* by declaring that μ presents the mean and $C^{T}C$ the Cholesky decomposition of the covariance matrix. Further we put
(7)$$\sigma_{i}=c_{ii},\;r_{ij}=\frac{c_{ij}}{c_{ii}}\;\left(i,j\in \left\{1,\cdots ,n\right\}\right),\;C=\mathrm{diag}\left(\sigma \right)\left[r_{ij}\right].$$

The matrix $\left[r_{ij}\right]$ is unitriangular, i.e., a triangular matrix whose diagonal entries are all 1. Then, each point $x=\left(\mu ,\sigma ,r\right)\in U=R^{n}\times {\left(R_{>0}\right)}^{n}\times R^{n(n-1)/2}$ presents the volume form
(8)$$V_{x}=\frac{1}{\sqrt{{\left(2\pi \right)}^{n}}\sigma_{1}\cdots \sigma_{n}}\exp\left(-\frac{1}{2}{∥C{\left(\sigma ,r\right)}^{-T}\left(y-\mu \right)∥}^{2}\right)dvol.$$

Let ${∥\cdot ∥}^{2}$ denote the sum of the squares of the entries of a matrix as well as a vector. The relative entropy defines the premetric $D(x,x^{\prime }=\left(\mu^{\prime },\sigma^{\prime },r^{\prime }\right))$ by
(9)$$D\left(x,x^{\prime }\right)=\frac{{∥C{\left(\sigma^{\prime },r^{\prime }\right)}^{-T}\left(\mu^{\prime }-\mu \right)∥}^{2}}{2}+\frac{∥C\left(\sigma ,r\right)C{\left(\sigma^{\prime },r^{\prime }\right)}^{-1}{∥}^{2}-n}{2}-\sum_{i=1}^{n}\ln\frac{\sigma_{i}}{\sigma_{i}^{\prime }},$$

Let $1_{n}$ denote the unit matrix, and ΔC the difference C(σ+Δσ,r+Δr)−C(σ,r). We have
(10)$$D\left(x+\Delta x,x\right)=\frac{{∥C^{-T}\Delta \mu ∥}^{2}}{2}+\frac{{∥\Delta CC^{-1}∥}^{2}}{2}+\mathrm{tr}\left(\Delta CC^{-1}\right)-\ln\left|1_{n}+\Delta CC^{-1}\right|.$$

Let $r^{ij}$ be the entries of the inverse matrix of $\left[r_{ij}\right]$. We have
(11)(theij-entryofΔCC−1)=Δσiσi(i=j)σi+Δσiσj∑k=i+1jrkjΔrik(i<j)110(i>j).

From Equations (Equation 10) and (Equation 11), we see that the Fisher information metric *g* is expressed as
(12)$$g=\sum_{k=1}^{n}{\left(\frac{1}{\sigma_{k}}\sum_{i=1}^{k}r^{ik}d\mu_{i}\right)}^{2}+2\sum_{i=1}^{n}{\left(\frac{d\sigma_{i}}{\sigma_{i}}\right)}^{2}+\sum_{l=1}^{n-1}\sum_{k=l+1}^{n}{\left(\frac{\sigma_{l}}{\sigma_{k}}\sum_{i=l+1}^{k}r^{ik}dr_{li}\right)}^{2}.$$

Put
(13)$$\begin{array}{cc} \; & g_{\mu \mu }=\left[g_{\mu_{i},\mu_{j}}\right]=\left[\sum_{k\geq i,j}\frac{r^{ik}r^{jk}}{{\sigma_{k}}^{2}}\right]=C^{-1}C^{-T},\\ \; & g_{\sigma \sigma }=\mathrm{diag}\left(\left(\frac{2}{{\sigma_{i}}^{2}}\right)\right),\\ \; & g_{rr,l}={\left[g_{r_{li}r_{lj}}\right]}_{i,j>l}={\left[{\sigma_{l}}^{2}g_{\mu_{i},\mu_{j}}\right]}_{i,j>l}\;\left(l=1,\cdots ,n-1\right). \end{array}$$

Then, the representing matrix of *g* is the following block diagonal matrix:(14)$$\mathrm{diag}(g_{\mu \mu },g_{\sigma \sigma },g_{rr,2},\cdots ,g_{rr,n}).$$

Lowering the upper indices of the α-connection with $\sum_{L}g_{KL}\Gamma^{\alpha }{{}_{IJ}}^{K}=\Gamma_{\{I,J\},K}^{\alpha }$, we have
(15)$$\begin{array}{cc} \; & \Gamma_{\left\{\mu_{i},\mu_{j}\right\},\sigma_{k}}^{0}=-\Gamma_{\left\{\mu_{i},\sigma_{k}\right\},\mu_{j}}^{0}=\frac{r^{ik}r^{jk}}{{\sigma_{k}}^{3}}, \end{array}$$
(16)$$\begin{array}{cc} \; & \Gamma_{\left\{\sigma_{i},\sigma_{i}\right\},\sigma_{i}}^{0}=\frac{-2}{{\sigma_{i}}^{3}}, \end{array}$$
(17)$$\begin{array}{cc} \; & \Gamma_{\left\{\mu_{i},\mu_{j}\right\},r_{ab}}^{0}=-\Gamma_{\left\{\mu_{i},r_{ab}\right\},\mu_{j}}^{0}=\sum_{k=b}^{n}\frac{r^{bk}\left(r^{ia}r^{jk}+r^{ik}r^{ja}\right)}{2{\sigma_{k}}^{2}}, \end{array}$$
(18)$$\begin{array}{cc} \; & \Gamma_{\left\{r_{li},r_{lj}\right\},\sigma_{l}}^{0}=-\Gamma_{\left\{r_{li},\sigma_{l}\right\},r_{lj}}^{0}=\sum_{k\geq i,j}\frac{-\sigma_{l}r^{ik}r^{jk}}{{\sigma_{k}}^{2}}, \end{array}$$
(19)$$\begin{array}{cc} \; & \Gamma_{\left\{r_{li},r_{lj}\right\},\sigma_{k}}^{0}=-\Gamma_{\left\{r_{li},\sigma_{k}\right\},r_{lj}}^{0}=\frac{{\sigma_{l}}^{2}r^{ik}r^{jk}}{{\sigma_{k}}^{3}}\;\left(k\geq i,j\right), \end{array}$$
(20)$$\begin{array}{cc} \; & \Gamma_{\left\{r_{li},r_{lj}\right\},r_{ab}}^{0}=-\Gamma_{\left\{r_{li},r_{ab}\right\},r_{lj}}^{0}={\sigma_{l}}^{2}\Gamma_{\left\{\mu_{i},\mu_{j}\right\},r_{ab}}^{0}\;\left(a>l\right), \end{array}$$
(21)$$\begin{array}{cc} \; & \Gamma_{\{I,J\},K}^{0}=0\;\left(\mathrm{for}\mathrm{the}\mathrm{other}\mathrm{choices}\mathrm{of}\;\{I,J\}\mathrm{and}K\right). \end{array}$$

With respect to the same coordinates, the coefficients of the e-connection are
(22)$$\begin{array}{cc} \; & \Gamma_{\left\{\mu_{i},\sigma_{k}\right\},\mu_{j}}^{1}=2\Gamma_{\left\{\mu_{i},\sigma_{k}\right\},\mu_{j}}^{0}, \end{array}$$
(23)$$\begin{array}{cc} \; & \Gamma_{\left\{\sigma_{i},\sigma_{i}\right\},\sigma_{i}}^{1}=3\Gamma_{\left\{\sigma_{i},\sigma_{i}\right\},\sigma_{i}}^{0}, \end{array}$$
(24)$$\begin{array}{cc} \; & \Gamma_{\left\{\mu_{i},r_{ab}\right\},\mu_{j}}^{1}=2\Gamma_{\left\{\mu_{i},r_{ab}\right\},\mu_{j}}^{0}, \end{array}$$
(25)$$\begin{array}{cc} \; & \Gamma_{\left\{r_{li},r_{l_{j}}\right\},\sigma_{l}}^{1}=2\Gamma_{\left\{r_{li},r_{l_{j}}\right\},\sigma_{l}}^{0}, \end{array}$$
(26)$$\begin{array}{cc} \; & \Gamma_{\left\{r_{li},\sigma_{k}\right\},r_{l_{j}}}^{1}=2\Gamma_{\left\{r_{li},\sigma_{k}\right\},r_{l_{j}}}^{0}\;\left(k\geq i,j\right), \end{array}$$
(27)$$\begin{array}{cc} \; & \Gamma_{\left\{r_{li},r_{ab}\right\},r_{lj}}^{1}=2\Gamma_{\left\{r_{li},r_{ab}\right\},r_{lj}}^{0}\;\left(a>l\right), \end{array}$$
(28)$$\begin{array}{c} \Gamma_{\{I,J\},K}^{1}=0\;\left(\mathrm{for}\mathrm{the}\mathrm{other}\mathrm{choices}\mathrm{of}\;\{I,J\}\mathrm{and}K\right). \end{array}$$

Those of the m-connection are
(29)$$\begin{array}{cc} \; & \Gamma_{\left\{\mu_{i},\mu_{j}\right\},\sigma_{k}}^{\left(-1\right)}=2\Gamma_{\left\{\mu_{i},\mu_{j}\right\},\sigma_{j}}^{0}, \end{array}$$
(30)$$\begin{array}{cc} \; & \Gamma_{\left\{\sigma_{i},\sigma_{i}\right\},\sigma_{i}}^{\left(-1\right)}=-\Gamma_{\left\{\sigma_{i},\sigma_{i}\right\},\sigma_{i}}^{0}, \end{array}$$
(31)$$\begin{array}{cc} \; & \Gamma_{\left\{\mu_{i},\mu_{j}\right\},r_{ab}}^{\left(-1\right)}=2\Gamma_{\left\{\mu_{i},\mu_{j}\right\},r_{ab}}^{0}, \end{array}$$
(32)$$\begin{array}{cc} \; & \Gamma_{\left\{r_{li},\sigma_{l}\right\},r_{l_{j}}}^{\left(-1\right)}=2\Gamma_{\left\{r_{li},\sigma_{l}\right\},r_{l_{j}}}^{0}, \end{array}$$
(33)$$\begin{array}{cc} \; & \Gamma_{\left\{r_{li},r_{l_{j}}\right\},\sigma_{k}}^{\left(-1\right)}=2\Gamma_{\left\{r_{li},r_{l_{j}}\right\},\sigma_{k}}^{0}\;\left(k\geq i,j\right), \end{array}$$
(34)$$\begin{array}{cc} \; & \Gamma_{\left\{r_{li},r_{lj}\right\},r_{ab}}^{\left(-1\right)}=2\Gamma_{\left\{r_{li},r_{lj}\right\},r_{ab}}^{0}\;\left(a>l\right), \end{array}$$
(35)$$\begin{array}{cc} \; & \Gamma_{\{I,J\},K}^{\left(-1\right)}=0\;\left(\mathrm{for}\mathrm{the}\mathrm{other}\mathrm{choices}\mathrm{of}\;\{I,J\}\mathrm{and}K\right). \end{array}$$
There is a particular system of coordinates for describing the e-connection. Namely, all the coefficients vanish with respect to the natural parameter $\left(C^{-1}C^{-T}\mu ,\xi \right)$, where $\xi ={\left(\xi_{ab}\right)}_{1\leq a\leq b\leq n}$ is the upper half of $C^{-1}C^{-T}$. On the other hand, all the coefficients for the m-connection vanish with respect to the expectation parameter (μ,ν), where $\nu ={\left(\nu_{ab}\right)}_{1\leq a\leq b\leq n}$ is the upper half of $C^{T}C+\mu \mu^{T}$.

### 2.5. The Generalization

This subsection is devoted to the generalization of the result of the author, which is mentioned in the introduction, to the above multivariate setting. We fix the third component *r* of the coordinate system (μ,σ,r), and change the presentation of the others. Namely, we take the natural projection $\pi :U=H^{n}\times R^{n(n-1)/2}\to R^{n(n-1)/2}$ and replace the coordinates (μ,σ) on the fiber $L\left(r\right)=\pi^{-1}\left(r\right)$ by (m,s) appearing in the next proposition. The generalization is then straightforward.

**Proposition** **1.**
*The fiber $L\left(r\right)=\pi^{-1}\left(r\right)$ is an affine subspace of U with respect to the e-connection $\nabla^{1}$. It can be parameterized by affine parameters $\frac{m_{i}}{{s_{i}}^{2}}$ and $\frac{1}{{s_{i}}^{2}}$, where $m={\left[r^{ij}\right]}^{T}\mu$ and $s=\sqrt{2}\sigma$.*


**Proof.** The natural parameters $C^{-1}C^{-T}\mu ={\left(2\sum_{k=i}^{n}r^{ik}\frac{m_{k}}{{s_{k}}^{2}}\right)}_{1\leq i\leq n}$ and $\xi ={\left(-\sum_{k=b}^{n}r^{ak}r^{bk}\frac{1}{{s_{k}}^{2}}\right)}_{1\leq a\leq b\leq n}$ are affine provided that $r^{ij}$ are constant, and $\frac{m_{i}}{{s_{i}}^{2}}$ and $\frac{1}{{s_{i}}^{2}}$ are affine. □

Note that $dr(\nabla_{\partial_{\mu }}^{\alpha }\partial_{\mu })$ is identically zero on some/any fiber L(r) if and only if α=1. The fiber satisfies the following two properties.

**Proposition** **2.**
*L(r) is closed under the convolution ∗ and the Bayesian product ·, and thus inherits them.*


**Proof.** The covariance of the density at $\left(\mu ,\sigma ,r\right)*\left(\mu^{\prime },\sigma^{\prime },r\right)$ is
(36)$${\left[r_{ij}\right]}^{T}\mathrm{diag}\left(\left({\sigma_{i}}^{2}\right)\right)\left[r_{ij}\right]+{\left[r_{ij}\right]}^{T}\mathrm{diag}\left(\left({\sigma_{i}^{\prime }}^{2}\right)\right)\left[r_{ij}\right]={\left[r_{ij}\right]}^{T}\mathrm{diag}\left(\left({\sigma_{i}}^{2}+{\sigma_{i}^{\prime }}^{2}\right)\right)\left[r_{ij}\right].$$This coincides with that of the density at $\left(\mu +\mu^{\prime },\left(\sqrt{{\sigma_{i}}^{2}+{\sigma_{i}^{\prime }}^{2}}\right),r\right)$. Thus L(r) is closed under ∗, and inherits it as $\left(m,s\right)*\left(m^{\prime },s^{\prime }\right)=\left(m+m^{\prime },\left(\sqrt{{s_{i}}^{2}+{s_{i}^{\prime }}^{2}}\right)\right).$Put $u=({\sigma_{i}}^{-2})$ and $y={\left[r^{ij}\right]}^{T}x$. The density at (μ,σ,r) is proportional to $\exp\left(y{\left(x\right)}^{T}\mathrm{diag}\left(u\right)y\left(x\right)-2m^{T}\mathrm{diag}\left(u\right)y\left(x\right)\right)$. From this we see that $\left(\mu ,\sigma ,r\right)\cdot \left(\mu^{\prime },\sigma^{\prime },r\right)=\left(\mu^{\prime \prime },\sigma^{\prime \prime },r^{\prime \prime }\right)$ implies $r^{\prime \prime }=r$, $u^{\prime \prime }=u+u^{\prime }$ and $\mathrm{diag}\left(u^{\prime \prime }\right)m^{\prime \prime }=\mathrm{diag}\left(u\right)m+\mathrm{diag}\left(u^{\prime }\right)$. Thus L(r) is closed under ·, and inherits it as $\left(m,s\right)\cdot \left(m^{\prime },s^{\prime }\right)=\left(\frac{m_{i}{s_{i}^{\prime }}^{2}+m_{i}^{\prime }{s_{i}}^{2}}{{s_{i}}^{2}+{s_{i}^{\prime }}^{2}},\frac{s_{i}s_{i}^{\prime }}{\sqrt{{s_{i}}^{2}+{s_{i}^{\prime }}^{2}}}\right)$. □

**Proposition** **3.**
*The fiber L(r) with the induced metric from g admits a Kähler complex structure.*


**Proof.** The restriction of *g* is $2\sum_{i=1}^{n}\frac{d{m_{i}}^{2}+d{s_{i}}^{2}}{{s_{i}}^{2}}$. We define the complex structure J:TL(r)→TL(r) by $J\left(\partial_{m_{i}}\right)=\partial_{s_{i}}$ ($\Leftrightarrow J\left(\partial_{s_{i}}\right)=-\partial_{m_{i}}$). Then, the 2-form $\omega =2\sum_{i=1}^{n}\frac{dm_{i}\wedge ds_{i}}{{s_{i}}^{2}}$ satisfies ω(·,J·)=g(·,·). □

We write the restriction ${D|}_{L(r)}$ of the premetric *D* using the coordinates (m,s) as follows, where we omit *r* for the expression does not depend on *r*.
(37)$${D|}_{L}\left(\left(m,s\right),\left(m^{\prime },s^{\prime }\right)\right)=\frac{1}{2}\sum_{i=1}^{n}\left\{{\left(\frac{m_{i}^{\prime }}{s_{i}^{\prime }}-\frac{m_{i}}{s_{i}^{\prime }}\right)}^{2}+\frac{{s_{i}}^{2}}{{s_{i}^{\prime }}^{2}}-1-\ln\frac{{s_{i}}^{2}}{{s_{i}^{\prime }}^{2}}\right\}.$$

We take the product $U_{1}\times U_{2}$ of two copies of *U*. Then, the products $L_{1}\left(r\right)\times L_{2}\left(R\right)$ of the fibers foliate $U_{1}\times U_{2}$. We call this the *primary* foliation of $U_{1}\times U_{2}$. For each $\left(r,R\right)\in R^{n(n-1)}$, we have the coordinate system (m,s,M,S) on the leaf $L_{1}\left(r\right)\times L_{2}\left(R\right)$. From the Kähler forms $\omega_{1}=2\sum_{i=1}^{n}\frac{dm_{i}\wedge ds_{i}}{{s_{i}}^{2}}$ and $\omega_{2}=2\sum_{i=1}^{n}\frac{dM_{i}\wedge dS_{i}}{{S_{i}}^{2}}$ respectively on $L_{1}\left(r\right)$ and $L_{2}\left(R\right)$, we define the symplectic forms $\omega_{1}\pm \omega_{2}$ on $L_{1}\left(r\right)\times L_{2}\left(R\right)$, which induce the mutually opposite orientations in the case where *n* is odd. Hereafter, we consider the pair of regular Poisson structures defined by these symplectic structures on the primary foliation, and fix the primitive 1-forms $\lambda_{\pm }=2\sum_{i=1}^{n}\left(\frac{dm_{i}}{s_{i}}\pm \frac{dM_{i}}{S_{i}}\right)$. The corresponding pair of Poisson bi-vectors is $\Pi_{\pm }=\frac{1}{2}\sum_{i=1}^{n}\left({s_{i}}^{2}\partial_{m_{i}}\wedge \partial_{s_{i}}\pm {S_{i}}^{2}\partial_{M_{i}}\wedge \partial_{S_{i}}\right)$ defined on $U_{1}\times U_{2}$. We take the 2n-dimensional submanifolds $F_{\varepsilon ,\delta }=\left\{\frac{m_{i}}{s_{i}}+\frac{M_{i}-\varepsilon_{i}}{S_{i}}=0,\;s_{i}S_{i}=\delta_{i}\;\left(i=1,\cdots ,n\right)\right\}$ of the leaf $L_{1}\left(r\right)\times L_{2}\left(R\right)$ for $\varepsilon \in R^{n}$ and $\delta \in {\left(R_{>0}\right)}^{n}$. The *secondary* foliation of $U_{1}\times U_{2}$ foliates any leaf U(r)×U(r) by the 3n-dimensional submanifolds $F_{\varepsilon }=\underset{\delta \in {\left(R_{>0}\right)}^{n}}{⋃}F_{\varepsilon ,\delta }$ for $\varepsilon \in R^{n}$. The *tertiary* foliation of $U_{1}\times U_{2}$ foliates all leaves $F_{\varepsilon }$ of the secondary foliation by the 2n-dimensional submanifolds $F_{\varepsilon ,\delta }$ for $\delta \in {\left(R_{>0}\right)}^{n}$.

**Proposition** **4.**
*With respect to the Kähler form $d\lambda_{-}$, the tertiary leaves $F_{\varepsilon ,\delta }$ are Lagrangian correspondences.*


**Proof.** The tangent space $TF_{\varepsilon ,\delta }$ is generated by $s_{i}\partial_{m_{i}}-S_{i}\partial_{M_{i}}$ and $2m_{i}\partial_{m_{i}}+s_{i}\partial_{s_{i}}-S_{i}\partial_{S_{i}}$. We have $d\lambda_{-}\left(s_{i}\partial_{m_{i}}-S_{i}\partial_{M_{i}},\partial_{m_{i}}\right)=0$ and $d\lambda_{-}\left(s_{i}\partial_{m_{i}}-S_{i}\partial_{M_{i}},s_{i}\partial_{s_{i}}-S_{i}\partial_{S_{i}}\right)=2\left(\frac{{s_{i}}^{2}}{{s_{i}}^{2}}-\frac{{\left(-S_{i}\right)}^{2}}{{S_{i}}^{2}}\right)=0$. Thus $F_{\varepsilon ,\delta }$ are Lagrangian submanifolds of $\left(L_{1}\left(r\right)\times L_{2}\left(R\right),d\lambda_{-}\right)$. □

The restrictions of $\lambda_{\pm }{|}_{N}$ to the hypersurface $N=\left\{\left(m,s,M,S\right)\in L_{1}\times L_{2}\;|\;\prod_{i=1}^{n}\left(s_{i}S_{i}\right)=1\right\}$ are contact forms. Let $\eta_{\pm }$ denote them.

**Proposition** **5.**
*For any ε and δ with $\prod_{i=1}^{n}\delta_{i}=1$, the submanifold $F_{\varepsilon ,\delta }\subset N$ is a disjoint union of n-dimensional submanifolds $\left\{s=const\right\}\subset F_{\varepsilon ,\delta }$ which are integral submanifolds of the hyper plane field $\ker\eta_{+}$ on N.*


**Proof.** We have $\lambda_{+}\left(s_{i}\partial_{m_{i}}-S_{i}\partial_{M_{i}}\right)=1-1=0$. □

For each point $\left(\varepsilon ,\delta \right)\in H^{n}$, we have the diffeomorphism ${\hat{F}}_{\varepsilon ,\delta }:H^{n}\to H^{n}$ sending $\left(m^{\prime },s^{\prime }\right)\in H^{n}$ to $\left(M,S\right)\in H^{n}$ with $\left(m^{\prime },s^{\prime },M,S\right)\in F_{\varepsilon ,\delta }$. We put $h_{i}=-\ln\frac{s_{i}S_{i}}{\delta_{i}}$ and
(38)$$f_{\varepsilon ,\delta }\left(m,s,M,S\right)=\frac{1}{2}\sum_{i=1}^{n}\left\{{\left(\frac{M_{i}-\varepsilon_{i}}{S_{i}}+e^{-h_{i}}\frac{m_{i}}{s_{i}}\right)}^{2}+e^{-2h_{i}}-1+2h_{i}\right\}.$$

Then, we have ${D|}_{L}\left(\left(m,s\right),\left(m^{\prime },s^{\prime }\right)\right)=f_{\varepsilon ,\delta }\left(\left(m,s\right),{\hat{F}}_{\varepsilon ,\delta }\left(m^{\prime },s^{\prime }\right)\right)$. For any $\zeta \in {\left(R_{>0}\right)}^{2n}$, we define the diffeomorphism $\varphi_{\varepsilon ,\zeta }:\left(m,s,M,S\right)\mapsto \left(\zeta_{2i-1}m_{i}\right),\left(\zeta_{2i-1}s_{i}\right),\left(\varepsilon_{i}+\zeta_{2i}\left(M_{i}-\varepsilon_{i}\right)\right),\left(\zeta_{2i}S_{i}\right)),$ which preserves the 1-forms $\lambda_{\pm }$. It is easy to prove the following proposition.

**Proposition** **6.**
*In the case where $\zeta_{2i-1}\zeta_{2i}=1$ for i=1,⋯,n, the diffeomorphism $\varphi_{\varepsilon ,\zeta }$ preserves $f_{\varepsilon ,\delta }$.*


For each $\varepsilon \in R^{n}$, we take the set $f_{\varepsilon }=\left\{\left(f_{\varepsilon ,\delta },F_{\varepsilon ,\delta }\right)|\delta \in {\left(R_{>0}\right)}^{n}\right\}$, and consider it as a structure of the secondary leaf $F_{\varepsilon }$.

**Proposition** **7.**
*For any $\zeta \in {\left(R_{>0}\right)}^{n}$, the diffeomorphism $\varphi_{\varepsilon ,\zeta }$ preserves the set $f_{\varepsilon }$ for any $\varepsilon \in R^{n}$. In the case where ζ satisfies $\prod_{i=1}^{n}\left(\zeta_{2i-1}\zeta_{2i}\right)=1$, the diffeomorphism $\varphi_{\varepsilon ,\zeta }$ also preserves the hypersurface N.*


**Proof.** $F_{\varepsilon ,\delta }$ maps to $F_{\varepsilon ,(\zeta_{2i-1}\zeta_{2i}\delta_{i})}$, and $f_{\varepsilon ,(\zeta_{2i-1}\zeta_{2i}\delta_{i})}\circ \varphi_{\varepsilon ,\zeta }=f_{\varepsilon ,\delta }$ holds. Further $\prod_{i=1}^{n}\delta_{i}=1$ implies $\prod_{i=1}^{n}\left(\zeta_{2i-1}\zeta_{2i}\delta_{i}\right)=1$ provided that $\prod_{i=1}^{n}\left(\zeta_{2i-1}\zeta_{2i}\right)=1$. □

For any $\delta \in {\left(R_{>0}\right)}^{n}$, the diffeomorphism ${\hat{F}}_{0,\delta }$ interchanges the operation $\left(m,s\right)*\left(m^{\prime },s^{\prime }\right)=\left(m+m^{\prime },\left(\sqrt{{s_{i}}^{2}+{s_{i}^{\prime }}^{2}}\right)\right)$ with the operation $\left(m,s\right)\cdot \left(m^{\prime },s^{\prime }\right)=\left(\frac{m_{i}{s_{i}^{\prime }}^{2}+m_{i}^{\prime }{s_{i}}^{2}}{{s_{i}}^{2}+{s_{i}^{\prime }}^{2}},\frac{s_{i}s_{i}}{\sqrt{{s_{i}}^{2}+{s_{i}^{\prime }}^{2}}}\right)$.

**Proposition** **8.**
*If $\left(m,s,M,S\right),\left(m^{\prime },s^{\prime },M^{\prime },S^{\prime }\right)\in F_{0,\delta }$, then*
(39)$$\left\{\begin{array}{cc} \left(\left(m,s\right)\cdot \left(m^{\prime },s^{\prime }\right),\left(M,S\right)*\left(M^{\prime },S^{\prime }\right)\right) & \in F_{0,\delta }\\ \left(\left(m,s\right)*\left(m^{\prime },s^{\prime }\right),\left(M,S\right)\cdot \left(M^{\prime },S^{\prime }\right)\right) & \in F_{0,\delta } \end{array}\right..$$


**Proof.** Putting $\left(m^{\prime \prime },s^{\prime \prime }\right)=\left(m,s\right)\cdot \left(m^{\prime },s^{\prime }\right)$ we have $M_{i}+M_{i}^{\prime }=-\frac{\delta_{i}m_{i}}{{s_{i}}^{2}}-\frac{\delta_{i}m_{i}^{\prime }}{{s_{i}^{\prime }}^{2}}=-\frac{\delta_{i}m_{i}^{\prime \prime }}{{s_{i}^{\prime \prime }}^{2}}$ and ${S_{i}}^{2}+{S_{i}^{\prime }}^{2}=\frac{{\delta_{i}}^{2}}{{s_{i}}^{2}}+\frac{{\delta_{i}}^{2}}{{s_{i}^{\prime }}^{2}}=\frac{{\delta_{i}}^{2}}{{s_{i}^{\prime \prime }}^{2}}$, hence the first equation. The second equation is similarly proved. □

A curve $\left(m\left(t\right),s\left(t\right)\right)\in H^{n}$ is a geodesic with respect to the e-connection $\nabla^{1}$ if and only if $\frac{m_{i}}{{s_{i}}^{2}}$ and $\frac{1}{{s_{i}}^{2}}$ are affine functions of *t* for i=1,⋯,n.

**Definition** **1.**
*We say that an e-geodesic $\left(m\left(t\right),s\left(t\right)\right)\in H^{n}$ is intensive if it admits an affine parameterization such that $\frac{1}{{s_{i}}^{2}}$ are linear for i=1,⋯,n.*


Note that any e-geodesic is intensive in the case where n=1.

**Proposition** **9.**
*Given an intensive e-geodesic (m(t),s(t)), we can change the parameterization of its image $\left(M(t),S(t)\right)=\left(\left(\varepsilon_{i}-\frac{m_{i}\left(t\right)\delta_{i}}{{s_{i}}^{2}}\right),\left(\frac{\delta_{i}}{s_{i}}\right)\right)$ under the diffeomorphism ${\hat{F}}_{\varepsilon ,\delta }$ to obtain an intensive e-geodesic.*


**Proof.** Put $\frac{m_{i}}{{s_{i}}^{2}}=a_{i}t+b_{i}$ and $\frac{1}{{s_{i}}^{2}}=c_{i}t$. We have $\frac{M_{i}-\varepsilon }{{S_{i}}^{2}}=-\frac{a_{i}}{c_{i}\delta_{i}}-\frac{b_{i}}{c_{i}\delta_{i}}\cdot \frac{1}{t}$ and $\frac{1}{{S_{i}}^{2}}=\frac{1}{c_{i}{\delta_{i}}^{2}}\cdot \frac{1}{t}$. They are respectively an affine function and a linear function of $\frac{1}{t}$. □

We have the hypersurface $N=\left\{\prod_{i=1}^{n}s_{i}S_{i}=1\right\}\subset H^{n}$ which carries the contact forms $\eta_{\pm }=2\sum_{i=1}^{n}\left(\frac{dm_{i}}{s_{i}}\pm \frac{dM_{i}}{S_{i}}\right){|}_{N}$. This is defined on any leaf $L_{1}\left(r\right)\times L_{2}\left(R\right)\approx H^{2n}$ of the primary foliation of $U_{1}\times U_{2}$. Now we state the main result.

**Theorem** **1.**
*The contact Hamiltonian vector field Y of the restriction of the function $\sum_{i=1}^{n}\frac{m_{i}}{s_{i}}$ to N for the contact form $\eta_{+}$ coincides with that for the other contact form $\eta_{-}$. It is tangent to the tertiary leaves $F_{\varepsilon ,\delta }$ and defines flows on them. Here each flow line presents the correspondence between intensive e-geodesics in Proposition 9.*


**Corollary** **1.**
*For any $\delta \in {\left(R_{>0}\right)}^{n}$, the flow on the leaf $F_{0,\delta }$ presents the iteration of the operation ∗ on the first factor of $U_{1}\times U_{2}$ and that of the operation · on the second factor as is described in Proposition 8 (see Figure 1).*


From Corollary 1, we see that the flow model of Bayesian inference studied in [4,5] also works for the multivariate case. We prove the theorem.

**Proof.** Take the vector field $\tilde{Y}=\frac{1}{4}\sum_{i=1}^{n}\left(2m_{i}\partial_{m_{i}}+s_{i}\partial_{s_{i}}-S_{i}\partial_{S_{i}}\right)$ on $U_{1}\times U_{1}$. It satisfies $\iota_{\tilde{Y}}\lambda_{\pm }=\sum_{i=1}^{n}\frac{m_{i}}{s_{i}}$, ${\tilde{L}}_{Y}\lambda_{\pm }=\lambda_{\pm }$, and $\iota_{\tilde{Y}}d\ln\left(s_{i}S_{i}\right)=0$, and thus its restriction to *N* is the contact Hamiltonian vector field *Y*. It also satisfies $\iota_{\tilde{Y}}d\left(\frac{m_{i}}{s_{i}}+\frac{M_{i}-\varepsilon_{i}}{S_{i}}\right)=\frac{1}{4}\left(\frac{m_{i}}{s_{i}}+\frac{M_{i}-\varepsilon_{i}}{S_{i}}\right)$ (i=1,⋯,n), where the right-hand sides vanish along $F_{\varepsilon ,\delta }$. Given a point $\left(P_{0},Q_{0}\right)=\left(m_{0},s_{0},r_{0},M_{0},S_{0},R_{0}\right)\in N\subset L_{1}\left(r_{0}\right)\times L_{2}\left(R_{0}\right)\subset U\times U$, we have the integral curve $\left(m\left(t\right),s\left(t\right),r\left(t\right),M\left(t\right),S\left(t\right),R\left(t\right)\right)=\left(e^{2t}m_{0},e^{t}s_{0},r_{0},M_{0},e^{-t}S_{0},R_{0}\right)$ of *Y* with initial point $\left(P_{0},Q_{0}\right)$. We can change the parameter of the curve $\left(m\left(t\right),s\left(t\right),r_{0}\right)$ on the first factor with $t^{\prime }=e^{-2t}$ to obtain an intensive e-geodesic. □

### 2.6. The Symmetry

The diffeomorphism $\varphi_{\varepsilon ,\zeta }$ with $\prod_{i=1}^{n}\left(\zeta_{2i-1}\zeta_{2i}\right)=1$ in Proposition 7 also appears in the standard construction of Hilbert modular cusps in [8]. We sketch the construction.

Since the function $H=\ln\prod_{i=1}^{n}\left(s_{i}S_{i}\right)$ is linear in the logarithmic space $R^{2n}∋\left(\left(\ln s_{i}\right),\left(\ln S_{i}\right)\right)$, we can take 2n−1 points $(\left(\ln\zeta_{2i-1}\left(k\right)\right),\left(\ln\zeta_{2i}\left(k\right)\right))\in \left\{H=0\right\}$ (k=1,⋯,2n−1) so that the quotient of the primary leaf L(r)×L(R) under the $Z^{2n-1}$-action generated by $\varphi_{\varepsilon ,\zeta (1)}$, ..., $\varphi_{\varepsilon ,\zeta (2n-1)}$ is the total space of a vector bundle over $T^{2n-1}\times R$. Here $T^{2n-1}$ is the quotient of {H=0} under the $Z^{2n-1}$-lattice generated by the above points, and the fiber $R^{2n}$ consists of the vectors (m,M−ε).

In the univariate case (n=1), on the logarithmic sS-plane, we can take any point $\left(\ln s,\ln S\right)=(\ln\zeta_{1}\left(1\right),\ln\zeta_{2}\left(1\right))$ of the line H=lns+lnS=0 other than the origin. That is, we put $\zeta_{1}\left(1\right)=e^{t}$ and $\zeta_{2}\left(1\right)=e^{-t}$ (t≠0). Then, the map $\varphi_{\varepsilon ,\zeta }$ sends (m,s,M,S) to $\left(e^{t}m,e^{t}s,e^{-t}\left(M-\varepsilon \right),e^{-t}S\right)$. The Z-action generated by it rolls up the level sets of *H*, so that the quotient of the logarithmic sS-plane becomes the cylinder $T^{1}\times R$, which is the base space. On the other hand, the mM-plane, which is the fiber, expands horizontally and contracts vertically. This is the inverse monodromy along $T^{1}$. In general, we obtain a similar $R^{2n}$-bundle over $T^{2n-1}\times R$ if we take the 2n−1 points of H=0 in general position.

From Proposition 7, we see that the leaf $F_{\varepsilon }$ of the secondary foliation with the set $f_{\varepsilon }$, as well as the pair of the 1-forms $\lambda_{\pm }$ with the function *H*, descends to the $R^{2n}$-bundle. If there exists further a $Z^{2n}$-lattice on the fiber $R^{2n}$ which is simultaneously preserved by the maps $\left(\left(m_{i}\right),\left(M_{i}-\varepsilon_{i}\right)\right)\mapsto \left(\left(\zeta_{2i-1}\left(k\right)m_{i}\right),\left(\zeta_{2i}\left(k\right)\left(M_{i}-\varepsilon_{i}\right)\right)\right)$ (k=1,⋯,2n−1), we obtain a $T^{2n}$-bundle over $T^{2n-1}\times R$. Such a choice of ζ(k) would be number theoretical. Indeed, this is the case for Hilbert modular cusps. Moreover, we are considering the 1-forms $\lambda_{\pm }$, which descend to the $T^{2n}$-bundle. See [9] for the standard construction with special attention to these 1-forms.

We should notice that the vector field *Y* does not descend to the $T^{2n}$-bundle. However, every actual Bayesian inference along *Y* eventually stops. Thus, we may take sufficiently large $T^{2n}$ and consider *Y* as a locally supported vector field to perform the inference in the quotient space.

## 3. Discussion

Finally, we would like to comment on the transverse geometry of the primary foliation. The author conjectures that it has some relation to the M-theory. See e.g., [10] for a relation between Poisson geometry and matrix theoretical or non-commutative geometrical physics.

The premetric D((μ+Δμ,σ+Δσ,r+Δr),(μ,σ,r)) on *U* can be decomposed as
(40)$$D\left(\left(\mu +\Delta \mu ,\sigma +\Delta \sigma ,r\right),\left(\mu ,\sigma ,r\right)\right)+\sum_{i=1}^{n-1}\sum_{j=i+1}^{n}{\left(\frac{\sigma_{i}+\Delta \sigma_{i}}{\sigma_{j}}\sum_{k=i+1}^{j}r^{kj}\Delta r_{ik}\right)}^{2}.$$

The first term on the right-hand side presents the fiber premetric ${D|}_{L}\left((m+\Delta m,s+\Delta s),(m,s)\right)$, where $\Delta m={\left[r^{ij}\right]}^{T}\Delta \mu$ and $\Delta s=\sqrt{2}\Delta \sigma$. If Δσ=0 (⇔Δs=0), then we have the (non-information geometrical) Pythagorean-type formula
(41)D((μ+Δμ,σ,r+Δr),(μ,σ,r))=D((μ+Δμ,σ,r),(μ,σ,r))+D((μ,σ,r+Δr),(μ,σ,r)).

Note that each term in this expression does not depend on μ. The second term on the right-hand side can be expressed as
(42)$$D\left(\left(\mu ,\sigma ,r+\Delta r\right),\left(\mu ,\sigma ,r\right)\right)=\sum_{i=1}^{n-1}\sum_{j=i+1}^{n}{\left(\frac{\sigma_{i}}{\sigma_{j}}\sum_{k=i+1}^{j}r^{kj}\Delta r_{ik}\right)}^{2}.$$

This presents the discretized version of the following restriction of the Fisher information *g*:(43)$${g|}_{\left\{(\mu ,\sigma )=\mathrm{const}\right\}}=\sum_{i=1}^{n-1}\sum_{j=i+1}^{n}{\left(\frac{\sigma_{i}}{\sigma_{j}}\sum_{k=i+1}^{j}r^{kj}dr_{ik}\right)}^{2}.$$

We have the orthonormal frame with respect to this metric which consists of
(44)$$e_{ij}=\frac{\sigma_{i}}{\sigma_{j}}\sum_{k=j}^{n}r_{jk}\partial_{r_{ik}}\;\left(1\leq i<j\leq n\right).$$

This frame satisfies the relations $\left[e_{ij},e_{kl}\right]=\delta_{il}e_{kj}-\delta_{kj}e_{il}$ of the unitriangular algebra, where $\delta_{\cdot ,\cdot }$ denotes Kronecker’s delta. Using the dual coframe $e^{ij}$, the relations can be expressed as
(45)$$de^{ij}=\sum_{k=i+1}^{j-1}e^{ik}\wedge e^{kj}.$$

The transverse section of the primary foliation of $U_{1}\times U_{2}$ is the product of two copies of the unitriangular Lie group, which we would like to call the bi-unitriangular group. We fix the frame (respectively, the coframe) of the transverse section consisting of the above $e_{ij}$ (respectively, $e^{ij}$) in the first factor $U_{1}$ and their copies $E_{ij}$ (respectively, $E^{ij}$) in the second factor $U_{2}$. The quotient manifold of the bi-unitriangular group by a cocompact lattice inherits a Riemannian metric from the sum of Fisher informations, and carries the following global (n−2)-plectic structure Ω (dΩ=0, $\Omega^{n}>0$):(46)$$\Omega =\sum_{i=1}^{n}e^{i,i+1}\wedge \cdots \wedge e^{i,n}\wedge E^{n-i+1,n-i+2}\wedge \cdots \wedge E^{n-i+1,n}.$$

We notice that, in the symplectic case where n=3, the quotient manifold admits no Kähler structure (see [11]). However, it is still remarkable that the transverse symplectic 6-manifold is naturally ignored in the Bayesian inference described in this paper. Conjecturally, a similar model would help us to treat events in parallel worlds (or blanes) in the same “psychological” procedure.

## Figures and Tables

**Figure 1 entropy-22-00240-f001:**
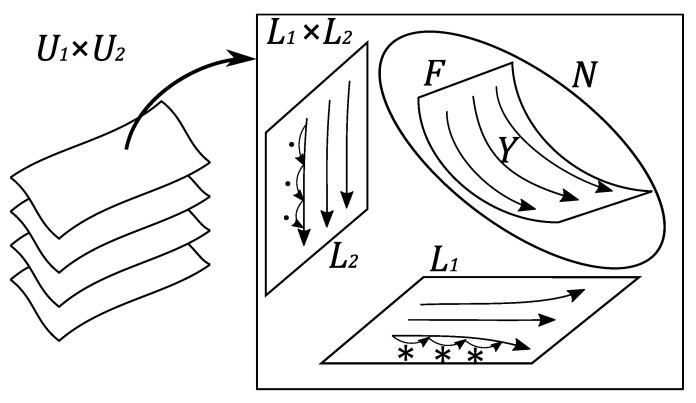
On any leaf of the primary foliation of $U_{1}\times U_{2}$, there is a bi-contact hypersurface *N* carrying the bi-contact Hamiltonian vector field *Y*. Because of the dimension, the surface *F* in the figure presents simultaneously a leaf of the secondary foliation and a leaf of the tertiary foliation of that leaf. The flow on the tertiary leaf $F=F_{0,\delta }$ traces the common lift of the iteration of ∗ on $L_{1}$ and the iteration of · on $L_{2}$.

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
