# Peer review of "Global Geometry of Bayesian Statistics"

_entropy, 2020, doi:10.3390/e22020240_

Round 1

Reviewer 1 Report

The paper "Global Geometry of Bayesian Statistics" by A. Mori deals with a kind of reformulation of information geometry in a language that would pretend to be more appealing for geometers and topologists. The key element employed to describe a family of differential geometric structures on a manifold U embedded in a space of probability distributions is a separating premetric, which is known in the field of information geometry under the name of "divergence function" or "contrast function". Although I appreciate the attempt to address information geometry in the community of topologists and geometers, I would point out that there is a huge literature about the connection between information geometry and geometry carried out through the concept of divergence function. Therefore I would recommend  the author to take into account that literature. Here follows some suggestions:

1)S. L. Lauritzen, Differential Geometry in Statistical Inference 1987
2)T. Kurose,"On the divergences of $1$-conformally flat statistical manifolds" 1994
3)S.Eguchi, Second Order Efficiency of Minimum Contrast Estimators in a Curved Exponential Family 1983
4)Ay et al., Information Geometry 2017
5)Felice&Ay, Towards a canonical divergence within information geometry, 2018

Author Response

The reviewer pointed out that many authors studied the connection between information geometry and geometry using divergence functions. Paying the respect to them, the author changed the abstract and the introduction. Particularly, he added the following two references suggested by the reviewer. 

4)Ay et al., Information Geometry 2017
5)Felice&Ay, Towards a canonical divergence within information geometry, 2018

Reviewer 2 Report

This paper extends the author's previous work [6] by investigating a foliation of the space of multivariate normal distributions, arriving from a special coordinate system. The result is given by theorem 1, further remarked by section 2.4, stating a global symmetry in terms of Bayesian learning and contact geometry.

Overall, I believe this work has enough novelty and deserves to be published. My main concern, however, is the clarity of the presentation, which can be greatly improved. Furthermore, there are some grammar issues especially in the non-math text, which can be improved by asking a college for proof-reading. Please see the following for detailed comments. I am putting a minor revision here because of the scientific soundness but expecting the authors can carefully go through the whole manuscript based on my comments.

Section 1 is too brief and not self-closed. Please introduce

the main results of [6] the expression of the two operations

Section 2, the introduction of Bayesian statistics. It is better to make explicit the relationship between (1) and Bayesian rule. In the beginning, mention that you introduce Bayesian learning from a perspective of agent collecting datum from a world, and what is the goal of the agent's learning. Overall it does not feel self-contained.

Section 2, it is better to add a sub-section of symplectic/contact geometry, and its relationship with Bayesian learning, as this may not be acquainted to the reader.

Page 5, put your derivations after line 95 into a separate section 2.4, stating at the beginning the outline and objectives of the derivations.

Theorem 1 is a bit wordy. Is it possible to make a shorter theorem plus a corollary?

Here the reader can benefit from an illustrative figure.

Section 2.4 the remarks are too brief and not intuitive. Please state the meaning and the intuition of theorem 1.

Can the author remark relationships with Bayesian information geometry, Hichem Snoussi?

Author Response

The reviewer concerned the clarity of the presentation of the manuscript. The author widely improved it and asked an expert to correct the grammar mistakes.

Section 1. According to the reviewer's comment, the author explained the previous result and included the expressions of the two operations.

Section 2. According to the reviewer's comment, the author made explicit the relationship between his updating formula and Bayesian rule. The author explained the aim of the learning practically and mathematically.

Section 3. According to the reviewer's comment, the author added a sub-section of symplectic/contact geometry and a sub-section of derivation elated to the main result. He also shorten Theorem 1 and add a corollary with an illustrative figure. The expression of the subsection of symmetry was improved. The author also added a comment on the work of Hichem Snoussi in the subsection of the information geometry.